# Vitamin D Receptor Gene Polymorphisms and Cigarette Smoking Impact on Oral Health: A Case-Control Study

**DOI:** 10.3390/ijerph17093192

**Published:** 2020-05-04

**Authors:** Aleksandra Suchanecka, Krzysztof Chmielowiec, Jolanta Chmielowiec, Grzegorz Trybek, Jolanta Masiak, Monika Michałowska-Sawczyn, Renata Nowicka, Katarzyna Grocholewicz, Anna Grzywacz

**Affiliations:** 1Independent Laboratory of Health Promotion, Pomeranian Medical University in Szczecin, 11 Chlapowskiego St., 70-204 Szczecin, Poland; o.suchanecka@gmail.com; 2Department of Hygiene and Epidemiology, Collegium Medicum, University of Zielona Góra, 28 Zyty St., 65-046 Zielona Góra, Poland; chmiele@vp.pl (K.C.); chmiele1@o2.pl (J.C.); 3Department of Oral Surgery, Pomeranian Medical University in Szczecin, 72 Powstanców Wlkp. St., 70-111 Szczecin, Poland; g.trybek@gmail.com; 4Neurophysiological Independent Unit, Department of Psychiatry, Medical University of Lublin, 20-093 Lublin, Poland; jolantamasiak@wp.pl; 5Faculty of Physical Culture, Gdańsk University of Physical Education and Sport, 80-853 Gdańsk, Poland; monikamichalowska@op.pl; 6109 Military Hospital with Cutpatient Cinic in Szczecin, 70-965 Szczecin, Poland; renatanowickaa@gmail.com; 7Department of Interdisciplinary Dentistry, Pomeranian Medical University, Al. Powstańców Wlkp. 72, 70-111 Szczecin, Poland; katarzyna.grocholewicz@pum.edu.pl

**Keywords:** vitamin D receptor gene, gingival index, oral health, polymorphism

## Abstract

Periodontal diseases are multiperspective problems resulting from numerous and diverse exposures that influence the process of initiation or progression of disease. The negative influence of tobacco smoking on oral health is well documented. The aim of the study was to analyze three SNPs in vitamin D receptor gene—rs7975232 (*ApaI*), rs2228570 (*FokI*) and rs1544410 (*BsmI*)—combined with oral health assessment—pH, gingival index, dry mouth, periodontitis, dry socket, redness of oral cavity mucosa, leukoplakia—in a group of cigarette smokers and in non-smokers. Moreover, the possibility of interactions between these polymorphisms and smoking was examined. When comparing the smokers and non-smokers groups, we noticed that rs1544410 heterozygotes were significantly more frequent in the first group, and for the second, both homozygotes were more frequent. Additionally, we observed the impact of interaction between the rs7975232 genotype and smoking status on gingival index. Current smoking was also associated with all analyzed oral health measures except for leucoplakia. Correlation between pH and age in both smokers and non-smokers was also present. Results of our analysis indicate that in our study group lifestyle and aging were leading factors associated with worse oral health status. However, the impact of genetic variants, and also the impact of their interaction with smoking on analyzed parameters was also visible. These results show great possibilities for all levels of prevention of oral diseases by means of education based on evidence-based medicine, but also for incorporating genetic testing and early interventions into this process for predisposed individuals.

## 1. Introduction

Periodontal diseases are multiperspective problems resulting from numerous and diverse exposures that influence the process of initiation or progression of disease. Particular importance is attached to several groups of factors, among them inherited (e.g., genetic variants), acquired such as social, educational and economic factors, and the local environment (e.g., biofilm load or composition), other diseases (e.g., sub-optimally controlled diabetes) and lifestyle (e.g., smoking, consumption of sugars, carbohydrate intake). The influence of these factors can be observed in a different combination in different individuals, and what is more they can induce differently weighted effects.

VDR (vitamin D receptor) is a nuclear receptor that binds to the active form of vitamin D. It affects various biological processes connected with bone metabolism and modulation of immune response [1]. VDR is included in a family of transcriptional regulatory factors with the sequence similar to the steroid and thyroid hormone receptors [2]. Vitamin D (1.25-dihydroxyvitamin D3) is a fat-soluble steroid hormone showing interaction with its nuclear receptor, vitamin D receptor, in the regulation of different biological processes, among them bone metabolism and immune response modulation. VDR influences the action of vitamin D by regulating its gene expression [3]. The *VDR* gene, which encodes vitamin D receptor, is composed of nine exons, spans approximately 75 kb, and is located on chromosome 12q13.11, with highest expression levels in the intestine, thyroid gland, and kidney [4]. In many studies, the authors have indicated the connection between gene polymorphism and immune functions, as well as bone or connective tissue metabolism, as they can be treated as the elements related to the pathogenesis of periodontal disease [5]. Epidemiological studies have suggested the existence of positive correlation between osteoporosis and alveolar bone and tooth loss, which indicates the fact that bad bone quality can be treated as a risk factor for chronic periodontitis (CP) [6,7]. Polymorphisms in the *VDR* gene are also considered to be associated with bone mineral density and as the effect of incidents of bone-related diseases, especially osteoporosis [8,9]. Moreover, there is a connection between *Mycobacterium tuberculosis* infections and chronic inflammatory diseases such as rheumatoid arthritis [10,11]. Hence, the hypothesis that *VDR* polymorphism may play an important role in susceptibility to poor oral health status seem to be well supported. Thus, far, several studies have analyzed the influence of the *VDR* polymorphisms rs2228570, rs1544410, rs7975232, and *TaqI* and their combinations on periodontitis [12,13]. However, they do not make it possible to draw unequivocal conclusions as to whether there exists a relationship between *VDR* polymorphisms and disease susceptibility. Additionally, only a few studies have concentrated on the interaction between previously mentioned polymorphisms and negative lifestyle factors, smoking among them, and the risk of periodontitis occurrence [14,15]. A meta-analysis of 15 studies demonstrated that *TaqI*, rs7975232, and rs1544410 polymorphisms in *VDR* presented an association with CP in the Asian populations; however, the same situation was not observed for Caucasians, and for both of those groups, no association was observed between rs2228570 polymorphism and CP [16]. A more recent meta-analysis including 18 studies made it possible to conclude that *TaqI* polymorphism was the only *VDR* polymorphism associated with CP in Asians [17]. The inconsistency of these results demonstrates the need to conduct more studies in various ethnic populations, which would make it possible to clarify the role of *VDR* polymorphism in CP. Polymorphisms rs7975232 and rs1544410 are located in the 8th intron of *VDR* gene, whereas the rs2228570 polymorphism is located in the second exon of the gene and creates a start codon, resulting in an alternative start site [18].

Tobacco smoking, along with hypertension and obesity, is considered the leading preventable cause of death in the world [19]. The negative influence of tobacco smoking on oral health is well documented. This includes both common and rare conditions, in the continuum from benign to life-threatening diseases. The most important among them are discoloration of teeth and dental restorations, bad breath, taste and smell disorders, impaired wound healing, periodontal disease, short-term and long-term implant, oral mucosal lesions such as smoker’s melanosis and smoker’s palate, potentially malignant lesions, and oral cancer [20]. Moreover, it has been noticed that cigarette smoking can modify the salivary system by increasing salivary fluid, with simultaneous reduction of salivary enzymes (i.e., amylase, lactic dehydrogenase, and acid phosphatase), and modification of anti-oxidizing enzymes (i.e., glutathione peroxidase) and immune system function [21,22].

Several studies have indicated that smoke exposure may be associated with caries, which results from alterations in saliva [23]. Confirmation of this fact may lie in the observation that children whose mothers smoked during their first 4 years of life suffer significantly more often from caries than children with non-smoking mothers. The use of tobacco, irrespective of the type of tobacco consumption, is a factor that increases the risk of periodontitis, and what is more, the risk is consistently connected with dosage of tobacco [23,24,25]. Health disorders connected with the oral cavity, including periodontium, are indicted with extremely complex etiology; hence, the consideration of both genetic and environmental factors is fundamental for holistic understanding of above-mentioned disease processes.

The aim of the study was to analyze three single nucleotide polymorphisms (SNPs) in vitamin D receptor gene (*VDR*)—rs7975232 (*Apa I*), rs2228570 *(Fok I)* and rs1544410 *(BsmI)*—combined with oral health assessment—pH, GI (gingival index), dry mouth, periodontitis, dry socket, redness of oral cavity mucosa, leukoplakia—in a group of cigarette smokers and in controls. Moreover, the possibility of interactions between these polymorphisms and smoking was examined. 

## 2. Materials and Methods 

### 2.1. Subjects

The study was conducted in the Independent Laboratory of Health Promotion, Pomeranian Medical University in Szczecin. Subjects were recruited in the Department of Oral Surgery, Pomeranian Medical University in Szczecin after obtaining the approval of the Bioethics Committee of the Pomeranian Medical University (KB-0012/164/17), as well as the informed, written consent of the subjects. The study group consisted of 400 people, among them 243 women (M = 33.16, SD = 13.1, minimal age = 20, maximal age = 78) and 157 men (M = 36.77, SD = 15.06, minimal age = 18, maximal age = 71). Oral health assessment—pH, GI (gingival index), dry mouth, periodontitis, dry socket, redness of oral cavity mucosa, leucoplakia was performed in all subjects divided into two groups—smokers and non-smokers. 

### 2.2. Laboratory Analysis

The genomic DNA was isolated from venous blood in compliance with standard procedures. The PCR method was used to genotype samples. All genotyping was performed with the fluorescence resonance energy transfer using the real-time PCR method on the LightCycler ^®^ 480 II System (Roche Diagnostic, Basel, Switzerland). For polymorphisms in the VDR gene, the following conditions were applied: PCR was performed with 50 ng DNA of each sample in a final volume of 20 µL containing 2 µL reaction mix, 0.5 mM of each primer, 0.2 mM of each hybridization probe, and 2 mM MgCl_2_ according to the manufacturer’s instructions with initial denaturation (95 °C for 10 min) and then 35 cycles of denaturation (95 °C for 10 s), annealing (60 °C for 10 s) and extension (72 °C for 15 s). After amplification, a melting curve was generated by holding the reaction at 40 °C for 20 s and then heating slowly to 95 °C. The fluorescence signal was plotted against temperature to provide melting curves for each sample. Peaks were obtained at 62.76 °C for the C allele and at 69.89 °C for the A allele of the rs7975232; at 58.55 °C for the A allele and at 66.30 °C for the G allele of the rs154441, and at 57.51 °C for the C allele and at 63.82 °C for the T allele of the rs2228570.

### 2.3. Statistical Analysis

The condition of homogeneity of variance was not accomplished (Levane test *p* ≤ 0.05); hence, for quantitative dependence variables (pH, GI), the U Mann-Whitney test was applied. Differences in frequencies in *VDR* genotypes between non-smokers and smokers were tested with chi-square test. The same test was applied to analyze the features characterizing oral cavity pathologies, such as dry mouth, periodontitis, dry socket, redness of oral cavity mucosa, leucoplakia. Moreover, Pearson’s linear correlation coefficient was calculated to show the relation between oral cavity pH for non-smoker and smoker groups. For association analysis between *VDR* genotypes and smoking and non-smoking, as well as oral pH and GI, multivariate analysis of factor effects ANOVA were used (pH/GI × genetic feature × non-smokers and smokers). Meanwhile, log linear analysis was used to show the association between *VDR* genotypes and smoking and non-smoking, as well as features of oral cavity pathologies. All calculations were performed using STATISTICA 13 (Tibco Software Inc., Palo Alto, CA, USA) for Windows (Microsoft Corporation, Redmond, WA, USA).

## 3. Results

The observed *VDR* rs1544410 and rs2228570 polymorphisms frequencies differed from expectations based on the Hardy-Weinberg theorem in the non-smokers group, but they did not differ in the smokers (Table 1).

In comparison with the control group, statistically significant differences in the genotypes frequency for the *VDR* rs1544410 gene were found in smokers (A/A 14% vs. A/A 18%, G/A 47.5% vs. G/A 35%, G/G 38.5% vs. G/G 0. 47%, χ2 =6.48, *p* = 0.039) (Table 2). There were no statistically significant differences in the allele frequency for the *VDR* rs1544410 between the smokers and the non-smokers (A 37.75% vs. A 35.5%, G 62.25% vs. G 64.5%, χ2 =0.42, *p* = 0.509) (Table 3). 

No differences were demonstrated in frequency of genes and alleles *VDR* rs2228570 and VDR rs797532 between smokers and non-smokers (Table 2 and Table 3). In addition, it was shown that oral cavity pH of smokers in comparison with non-smokers was statistically significantly decreased (6.45 vs. 6.90, Z = −5.276, *p* ≤ 0.000). We also demonstrated that GI is statistically significantly increased in case of smokers in comparison with non-smokers (1.15 vs. 0.45, Z = 8.062, *p* ≤ 0.000). Moreover, in the group of smokers we observed statistically significant higher frequency of mouth dryness (41.5% vs. 9%, χ2 = 56.83, *p* ≤ 0.000), periodontitis (42% vs. 6.5%, χ2 = 69.53, *p* ≤ 0.000), dry socket (15.5% vs. 3%, χ2 = 19.51, *p* ≤ 0.000), redness of oral cavity mucosa (33.5% vs. 6%%, χ2 = 48.62, *p* ≤ 0.000). Nevertheless, we did not observe statistically significant differences between smokers and non-smokers in case of leukoplakia occurrence (Table 4). Additionally, we noticed the correlation between age and pH of non-smokers group (r = −0.202, *p* = 0.004, Figure 1). A similar correlation between age and pH occurred in case of smokers (r = −0.354, *p* ≤ 0.000, Figure 2).

We did not notice any statistical differences in interactions in log-linear analysis in relation to frequency of *VDR* rs1544410, *VDR* rs2228570, *VDR* rs7975232 genotypes between smokers and non-smokers and pathologies in the oral cavity (mouth dryness, periodontitis, dry socket, redness of oral cavity mucosa, leukoplakia).

No statistical differences were noted in the case of interactions in variance analysis for arrangements of factors in connection with frequency of *VDR* rs1544410, *VDR* rs222857 genotypes between non-smokers and smokers, and pH, and GI. Additionally, an integration between frequency of *VDR* rs7975232 genotypes for non-smokers and smokers and pH-value was not found.

Statistical differences were noted in the case of interactions in variance analysis for arrangements of factors in connection with frequency of *VDR* rs7975232 genotypes between non-smokers and smokers, and GI (F(2, 394) = 4.102, *p* = 0.017, ɳ2 = 0.02, observed power = 0.726, Figure 3). Among the non-smoker group for polymorphism rs1544410 A/A of *VDR* gene GI was M = 0.382, SD = 0.402, A/C GI was M = 0.545, SD = 0.714, C/C GI was M = 0.387, SD = 0.527. Among the smoker group for polymorphism rs1544410 A/A of *VDR* gene GI was M = 1.310, SD = 0.954, A/C GI was M = 1.002, SD = 0.921, C/C GI was M = 1.256, SD = 0.852.

## 4. Discussion

Our study concentrated on the analysis of the association between *VDR* polymorphisms (rs7975232, rs1544410 and rs2228570) and oral health status, in connection with the influence of smoking. When comparing the smokers and non-smokers groups, we noticed that rs1544410 heterozygotes were significantly more frequent in the first group, and for the second, both homozygotes were more frequent. Additionally, we observed the impact of the interaction between the rs7975232 genotype and smoking status on gingival index (GI). What is more, current smoking was also associated with all analyzed oral health measures except for leucoplakia. Finally, there was also a correlation between pH and age in both smokers and controls.

As mentioned above, we found a significant impact of the interaction between rs7975232 polymorphism and smoking on the GI index. Meta-analysis of 30 studies suggested that this variant was significantly associated with the susceptibility to periodontitis under dominant genetic model in the overall population [26]. Further subgroup analyses yielded similar positive results for rs2228570 variants in East Asians and patients with chronic periodontitis. Nevertheless, no other positive findings were observed in overall and subgroup analyses. The authors concluded that rs2228570 variant might serve as a genetic biomarker of periodontitis [26].

We did not find any association with rs2228570 polymorphism, which is strongly associated with severe CP alone, and the combination of being genotype positive and smoking further increased the impact in Thai population [27]. This combined effect was 3.7 times greater than what would be expected from the sum of their individual effects, indicating a synergistic interaction. However, no association between rs2228570 polymorphism and CP was observed in the studies conducted among Japanese [28] and southern Chinese populations [13]. This discrepancy between studies may be related to ethnic differences in the distribution of *VDR* polymorphisms, with different variants being more frequent. Therefore, studies on different ethnic populations may yield different results. Nevertheless, results from the Thai population [27] indicated that none of the rs1544410, rs7975232, or *TaqI* haplotypes were significantly associated with CP. A similar finding was observed in a previous study in Japanese subjects [15]. However, different findings have been reported in other populations. The GT haplotype of rs1544410*-TaqI* polymorphisms was associated with CP in Brazilians [12]. In addition, the GAT haplotype of rs1544410, rs7975232, and *TaqI* polymorphisms increased susceptibility to severe CP in Turkish subjects [29]. It should be noted that rs1544410, rs7975232, and *TaqI* polymorphisms by themselves are not functional polymorphisms, but rather markers for unidentified functional alleles located elsewhere in the gene [1]. The functional alleles that are in linkage disequilibrium with these markers are likely to be different among ethnic groups. This could partly explain the varying results observed between studies.

Our results clearly indicate the impact of current smoking habits on measured oral health parameters. Especially GI index is of great importance, as it has been shown by other researchers that periodontitis risk dependent on tobacco consumption [23,24,25].

Our study shows a correlation between pH and age. Studies show a significant association between CP and age, which is in agreement with studies by Amarasena et al. [30] and Demetriou et al. [31]. It is suggested that as the length of time increases, the chronic plaque accumulation increases, followed by increase in periodontal tissues destruction [32]. This study also showed higher percentage of CP in men than in women. The causes of these sex differences have not been explained clearly, but suggest that men have poorer oral hygiene, less positive thinking toward oral health and dental-visit interaction, which is in agreement with a study by Nazish et al. [33]. On the other hand, women still have varied periodontal problems due to hormonal disturbance in various decades of life. In addition, not enough studies have been done in developing countries which might have different results as compared to developed countries. Other studies of hormonal effect on periodontal diseases suggested that estrogen hormone in females can affect tooth retention by preventing the alveolar bone resorption [34,35,36]. 17b-Estradiol, the primary female sex hormone, may also play a role in promoting periodontal regeneration in an experimental periodontitis model [37].

Our study is not free of some limitations. We analyzed only people of Caucasian origin; hence, the results ought to be verified in other populations. Our future analysis will incorporate methylation analysis on greater number of subjects to further analyze the subject in greater detail. However, regardless of the biological factor, we must pay special attention to the social factor, which is related to public health. Smoking has many side effects affecting health, including the oral cavity, but in the context of epidemiology, let this be another reason to quit smoking.

## 5. Conclusions

The results of our analysis indicate the influence of both genetic and lifestyle factors on the health of the oral cavity. We observed that active smoking was associated with decreased values of almost all analyzed parameters (except leukoplakia). Genetic analysis showed an association of rs1544410 with smoking status, but not with the quality of oral health. We also discovered the impact of interaction between rs7975232 variants and smoking status on GI. Additionally, correlation between pH and age in both smokers and non-smokers was also present. We conclude that in our study group, lifestyle and aging were the leading factors associated with worse oral health status. However, the impact of genetic variants, and the impact of their interaction with smoking on analyzed parameters was also visible. These results show great possibilities for all levels of prevention of oral diseases by means of education based on evidence-based medicine but also of incorporating to this process genetic testing and early interventions for predisposed individuals.

## Figures and Tables

**Figure 1 ijerph-17-03192-f001:**
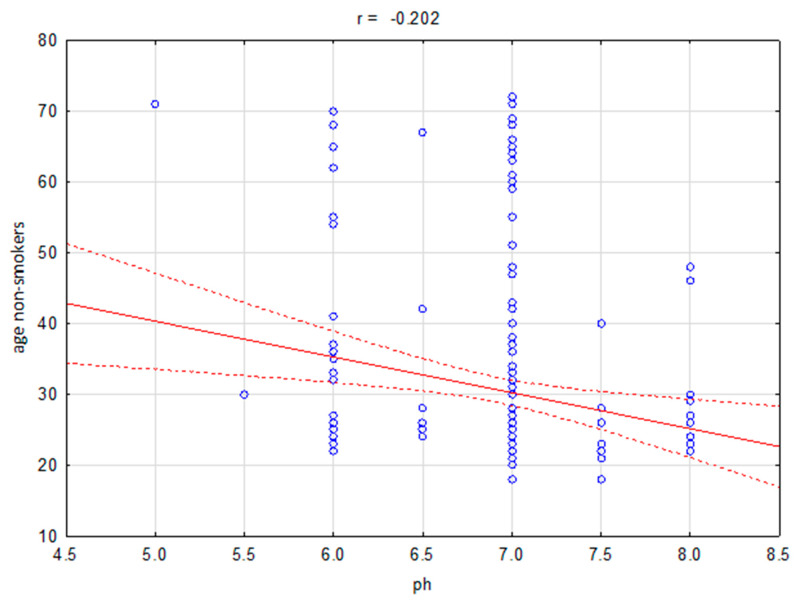
Correlation between age and pH for the non-smokers group (r = −0.202, *p* = 0.004).

**Figure 2 ijerph-17-03192-f002:**
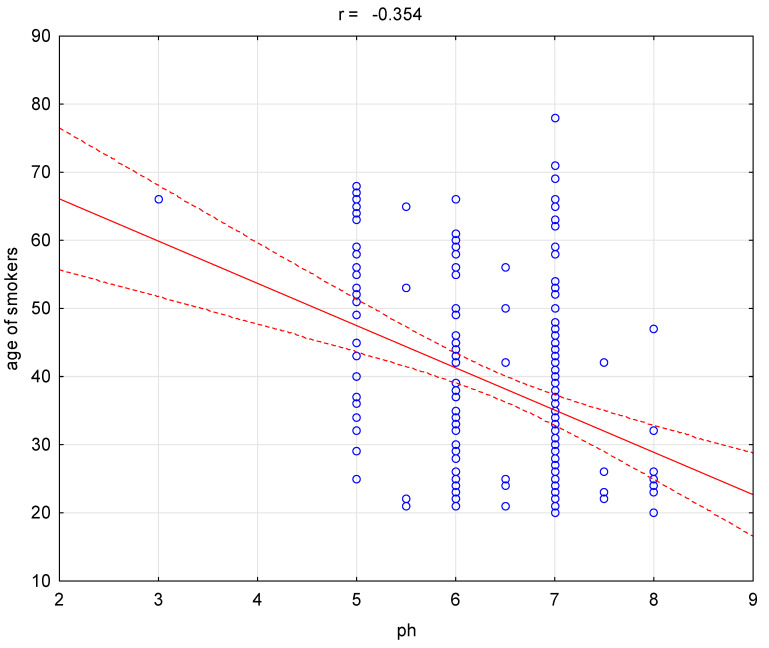
Correlation between age and pH for the smokers group (r = −0.354, *p* ≤ 0.000).

**Figure 3 ijerph-17-03192-f003:**
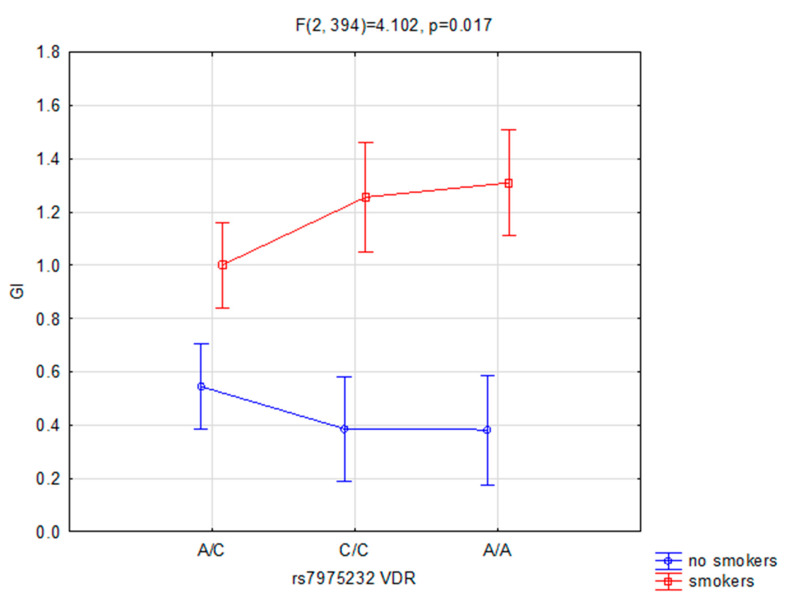
Interaction between smoking/non-smoking and rs7975232 and GI.

**Table 1 ijerph-17-03192-t001:** Hardy-Weinberg’s law for smokers and non-smokers.

Hardy-Weinberg Equilibrium Calculation Including Analysis for Ascertainment Bias		Observed (Expected)		Test χ^2^	
				χ^2^	*p*
***VDR* rs1544410**					
**Smokers** ***N* = 200**	A/A	28 (28.50)	G allele freq = 0.62 A allele freq = 0.38	0.02	>0.05
	G/A	95 (93.99)			
	G/G	77 (77.50)			
Non-Smokers *N* = 200	A/A	36 (25.205)	G allele freq = 0.65 A allele freq = 0.35	**11.11**	**<0.05**
	G/A	70 (91.59)			
	G/G	94 (83.20)			
***VDR* rs2228570**					
Smokers *N* = 200	T/T	38 (39.16)	C allele freq = 0.56 T allele freq = 0.44	0.11	>0.05
	C/T	101 (98.68)			
	C/C	61 (62.16)			
Non-Smokers *N* = 200	T/T	31 (39.60)	C allele freq = 0.55 A allele freq = 0.44	**6.06**	**<0.05**
	C/T	116 (98.79)			
	C/C	53 (61.60)			
***VDR* rs7975232**					
Smokers *N* = 200	A/A	57 (51.51)	C allele freq = 0.49 A allele freq = 0.51	2.41	>0.05
	A/C	89 (99.98)			
	C/C	54 (48.51)			
Non-Smokers *N* = 200	A/A	54 (47.53)	C allele freq = 0.51 A allele freq = 0.49	3.35	>0.05
	A/C	87 (99.94)			
	C/C	59 (52.53)			

*p*-statistical significance χ2 test, *N*—number of subjects, Significant between-group differences are marked in bold print.

**Table 2 ijerph-17-03192-t002:** Frequency of genotypes of the *VDR* rs1544410, *VDR* rs2228570, *VDR* rs7975232 genes polymorphisms in smokers and in non-smokers.

Frequency of Genotypes		*N* (%)	Test χ^2^	
			χ^2^	*p*
***VDR* rs1544410**				
**Smokers** ***N* = 200**	A/A	28 (14%)	**6.48**	**0.039**
	G/A	95 (47.5%)		
	G/G	77 (38.5%)		
Non-Smokers *N* = 200	A/A	36 (18%)		
	G/A	70 (35%)		
	G/G	94 (47%)		
***VDR* rs2228570**				
Smokers *N* = 200	T/T	38 (19%)	2.31	0.315
	C/T	101 (50.5%)		
	C/C	61 (31.5%)		
Non-Smokers *N* = 200	T/T	31 (15.5%)		
	C/T	116 (58%)		
	C/C	53 (26.5%)		
***VDR* rs7975232**				
Smokers *N* = 200	A/A	57 (28.5%)	0.325	0.850
	A/C	89 (44.5%)		
	C/C	54 (27%)		
Non-Smokers *N* = 200	A/A	54 (27%)		
	A/C	87 (43.5%)		
	C/C	59 (29.5%)		

*p*-statistical significance χ2 test, *N*—number of subjects, Significant between-group differences are marked in bold print.

**Table 3 ijerph-17-03192-t003:** Frequency of alleles of the *of the VDR* rs1544410, *VDR* rs2228570, *VDR* rs7975232 genes polymorphisms in smokers and in non-smokers.

Frequency of Alleles		*N* (%)	Test χ^2^	
			χ^2^	*p*
***VDR* rs1544410**				
**Smokers** ***N* = 200**	A	151 (37.75%)	0.42	0.509
	G	249 (62.25%)		
Non-Smokers *N* = 200	A	142 (35.5%)		
	G	258 (64.5%)		
***VDR* rs2228570**				
Smokers *N* = 200	T	177 (44.25%)	0.01	0.943
	C	223 (55.75)		
Non-Smokers *N* = 200	T	178 (44.5%)		
	C	222 (55.5%)		
***VDR* rs7975232**				
Smokers *N* = 200	A	203 (50.75%)	0.32	0.572
	C	197 (49.25%)		
Non-Smokers *N* = 200	A	195 (48.75)		
	C	205 (51.25)		

**Table 4 ijerph-17-03192-t004:** Oral health status parameters in smokers and non-smokers groups.

	Smokers (*N* = 200)	Non-Smokers (*N* = 200)	U Manna-Whitneya (Z) χ^2^	*p* Value
**pH**	6.45 ± 0.83	6.90 ± 0.52	**−5.276 ***	**≤0.000**
**GI**	1.15 ± 0.92	0.45 ± 0.59	**8.062 ***	**≤0.000**
**Mouth dryness**				
**No**	117 (58.5%)	182 (91%)	**56.833 ^#^**	**≤0.000**
**Yes**	83 (41.5%)	18 (9%)		
**Periodontitis**				
**No**	116 (58 %)	187 (93.5%)	**69.53 ^#^**	**≤0.000**
**Yes**	84 (42%)	13 (6.5%)		
**Dry socket**				
**No**	169 (84.5%)	194 (97%)	**19.51 ^#^**	**≤0.000**
**Yes**	31 (15.5%)	6 (3%)		
**Redness of oral cavity mucosa**				
**No**	133 (66.5%)	188 (94%)	**48.62 ^#^**	**≤0.000**
**Yes**	67 (33.5%)	12 (6%)		
**Leukoplakia**				
**No**	194 (97%)	198 (99%)	2.82	0.093
**Yes**	6 (3%)	2 (1%)		

*p*-statistical significance t-Student’s test, *N*—number of subjects. * test U Manna-Whitneya, ^#^ test Chi ^2^. Significant between-group differences are marked in bold print.

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
