# Peer review of "Vitamin D Receptor Gene Polymorphisms and Cigarette Smoking Impact on Oral Health: A Case-Control Study"

_ijerph, 2020, doi:10.3390/ijerph17093192_

Round 1
Reviewer 1 Report
Suchaneka et al. performed an observational and quantitative study on three SNPs on both alleles of the VDR gene. They tested whether these SNPs, smoking, age, and other parameters would be associated with oral health, specifically PD. Although the study is well designed and well executed, there are some issues with the presentation of the writing.
The introduction has the correct content, but the grammar and information flow needs to be polished. Here are some examples:
line 56: remove the word "success"
line 71-72: Rephrase the sentence to "Epidemiological studies have suggested the existence of a positive correlation between..." This sentence does not fit very well in the paragraph as is now. I suggest moving it further down where the ideas build up to it.
line 80: The sentence that starts with "VDR is a nuclear receptor..." should be the first sentence of this paragraph. You should describe the receptor, what it binds, what polymorphisms have been found, how these correlate with any diseases, what is the epidemiology of that and the rationale for your hypothesis. This information flow makes the paragraph more understandable to the non-specialist.
line 84: Rephrase "Moreover, there is a connection between Mycobacterium tuberculosis infections and chronic inflammatory diseases..."
line 85: bacteria scientific name in italics.
line 87: Rephrase "Thus far, several studies have analyzed..."
Throughout the whole paper, call one group "smokers" and the other group "non-smokers" and stick to this.
In the Results section, the quality of Figure 3 could be increased. It is blurry as is.
line 195: delete the word "While"
The results seem (mostly) non-significant to me. I do not see much of a correlation between the three SNPs tested and the other parameters. Perhaps this is because of the way the data is being presented, or the way the statistical analyses were done. I find it hard to follow and agree with the conclusions by the authors. If the data was a lot easier to digest and interpret, perhaps I would be arriving at the same conclusions, but not at the moment. I urge the authors to rethink how to present the data so that it best supports their current conclusions.
Author Response
ANSWER
Dear Reviewer,
We would like to thank you for your valuable comments on the article. Below you will find our reply to your review. All changes are with a description or a comment and changes have been made to the manuscript (track changes in the tracking group on the review tab).
Reviewer 1
Comments and Suggestions for Authors
Suchaneka et al. performed an observational and quantitative study on three SNPs on both alleles of the VDR gene. They tested whether these SNPs, smoking, age, and other parameters would be associated with oral health, specifically PD. Although the study is well designed and well executed, there are some issues with the presentation of the writing. The introduction has the correct content, but the grammar and information flow need to be polished. Here are some examples:
line 56: remove the word "success"
Thank you for this suggestion – we removed word “success” – line 60.
line 71-72: Rephrase the sentence to "Epidemiological studies have suggested the existence of a positive correlation between..." This sentence does not fit very well in the paragraph as is now. I suggest moving it further down where the ideas build up to it.
Thank you for this suggestion – the sentence has been rephrased and moved to lines 76-79.
line 80: The sentence that starts with "VDR is a nuclear receptor..." should be the first sentence of this paragraph. You should describe the receptor, what it binds, what polymorphisms have been found, how these correlates with any diseases, what is the epidemiology of that and the rationale for your hypothesis. This information flow makes the paragraph more understandable to the non-specialist.
Thank you for this comment – the paragraph has been changed – starting from line 65.
line 84: Rephrase "Moreover, there is a connection between Mycobacterium tuberculosis infections and chronic inflammatory diseases..."
Thank you for this suggestion – the sentence has been rephrased – line 80 - 82.
line 85: bacteria scientific name in italics.
Thank you for this catch – the proper formatting has been added – line 81.
line 87: Rephrase "Thus far, several studies have analyzed..."
Thank you for this suggestion – the sentence has been rephrased – lines 80-82.
Throughout the whole paper, call one group "smokers" and the other group "non-smokers" and stick to this.
Thank you for this suggestion – the names of both groups have been changed throughout the paper.
In the Results section, the quality of Figure 3 could be increased. It is blurry as is.
Thank you for this valuable comment – the quality of all 3 figures has been increased.
line 195: delete the word "While"
Thank you for this suggestion – we removed word “while” – line 192.
The results seem (mostly) non-significant to me. I do not see much of a correlation between the three SNPs tested and the other parameters. Perhaps this is because of the way the data is being presented, or the way the statistical analyses were done. I find it hard to follow and agree with the conclusions by the authors. If the data was a lot easier to digest and interpret, perhaps I would be arriving at the same conclusions, but not at the moment. I urge the authors to rethink how to present the data so that it best supports their current conclusions.
Thank you for this suggestion – after studying the data once more we concluded that our conclusions might not fit perfectly the results, and we changed these part of paper – lines 37- 42 and 263-273.
Reviewer 2 Report
This study is interesting, the authors evaluated the relationship between SNPs of vitamin D receptor gene (VDR) – rs7975232 111 (ApaI), rs2228570 (FokI) and rs1544410 (BsmI), in smoker and non-smoker patients. However, they include several variables related to smoking habits in special with the periodontitis. I think this study has some mistakes that must be resolved by the authors. Is it possible that authors describe the VDR gene SNPs according to: SNP id, primer used, restriction enzyme, localization, sequence position, base change and genotype, in a table? Is it possible that authors divide the variables between gender and relate them with the polymorphisms? Would it be possible to make a table with the clinical groups including: Gender, Age, Smoking habit, and associated pathologies? why did not the authors collect saliva samples of each patient? Which types of leukoplakia were present in the study? Was the mouth dryness only related with the smoking habit? Or did they have another important variables related to it?Author Response
ANSWER
Dear Reviewer,
We would like to thank you for your valuable comments on the article. Below you will find our reply to your review. All changes are with a description or a comment and changes have been made to the manuscript (track changes in the tracking group on the review tab).
Reviewer 2
Comments and Suggestions for Authors
This study is interesting, the authors evaluated the relationship between SNPs of vitamin D receptor gene (VDR) – rs7975232 111 (ApaI), rs2228570 (FokI) and rs1544410 (BsmI), in smoker and non-smoker patients. However, they include several variables related to smoking habits in special with the periodontitis. I think this study has some mistakes that must be resolved by the authors.
Is it possible that authors describe the VDR gene SNPs according to: SNP id, primer used, restriction enzyme, localization, sequence position, base change, and genotype, in a table?
Thank you for this suggestion. Our task is to place as much detail as possible in our research. In the case of this analysis we used real-time PCR melting-curve genotyping method, we did not use restriction enzymes – as described in the Methods section. However, we understand a possibility of confusion of a Reader, as we decided to use both rs numbers and classical names of polymorphic sites based on restriction enzymes. The reason for that was the fact that most Authors use either rs or enzyme names of SNPs analysed in this study. We wanted our paper to be as clear as possible in terms of analysed polymorphic sites. The location of gene in the chromosome as well as locations of SNPs within a gene has been given in Introduction.
Is it possible that authors divide the variables between gender and relate them with the polymorphisms?
Thank you for this suggestion. This was a matter of long debate and cautious consideration in our team. We publish this study as first of a series considering the topic of oral health in smokers. In the present study we did not want to show separate analysis for both sexes due to rather small number of subjects in each homogenous subgroup, which would strongly decrease scientific value of our research. The number of subjects that will be additionally recruited to our study group is considerably larger, and we hope that in the near future analysis incorporating division into homogeneous subgroups, e.g. separate analysis for men and women will be presented. However, we strongly agree with this comment, because our research methodology and designs are consistent with the usage homogenous subgroups.
Would it be possible to make a table with the clinical groups including: Gender, Age, Smoking habit, and associated pathologies?
why did not the authors collect saliva samples of each patient?
Which types of leukoplakia were present in the study?
Was the mouth dryness only related with the smoking habit? Or did they have another important variables related to it?
Thank you for these fundamental and very proper questions. In our study group leucoplakia was described as white or greyish patches on oral mucosa – we did not have cases of erytroplakia, speckled or hairy leucoplakia. As for the rest of clinical and demographic data – we did not want to construct too much small homogenous groups for reasons described in detail in previous answer.
We wanted to focus on case-control analysis showing possible associations. The project is still ongoing, apart from a temporary break in recruitment (due to pandemic) we invited two more Dentists to our team (there was only one). We think that further analyses on larger groups incorporating all clinical data, a thorough interview with the patient and their examination (and subsequent analysis of homogenous subgroups) - will significantly enrich the insight into the analysis and broaden our horizons. In addition, we are planning methylation and haplotype analysis.
Reviewer 3 Report
Comments and Suggestions for Authors:
In this manuscript, the authors investigate different SNPs in vitamin D receptor gene (VDR) in relation to the health of oral cavity of smokers and non-smokers subjects. The results confirm the involvement of genetic and smoke habits in oral health.
This document can be accepted although some points need to be clarified:
1) Clarify the methodology used in the analysis of samples and results, there are some shortcomings in the descriptions.
2) Please describe more in detail the specific components included in the tables 1-4, in order to help understanding the sentence included in the text.
3) Add a reference for sentence:
- line 218: Meta-analysis of 30 studies suggested that this variant was significantly associated with the susceptibility to periodontitis under dominant genetic model in the overall population.
4) Contextualize, more clearly, people origin differences, highlighted in other studies.
5) check the formatting of the text, and I suggest enter the name of the SNPs in the same way (or rs7975232 or ApaI, rs1544410 or BsmI, and rs2228570 or FokI)
6) “Periodontal disease is unconditionally associated with care for oral hygiene and the use of various types of psychoactive substances, especially inhaled - which includes smoking, among others. Such addiction affects not only the health of the oral cavity, but also the transmission of diseases by droplet, important in the light of recent events associated with the COVID-19/ SARS-CoV pandemic.” the sentence seems detached from the rest of the text, could be eliminated or should be inserted in a context and with specific reference.
Author Response
ANSWER
Dear Reviewer,
We would like to thank you for your valuable comments on the article. Below you will find our reply to your review. All changes are with a description or a comment and changes have been made to the manuscript (track changes in the tracking group on the review tab).
Reviewer 3
Comments and Suggestions for Authors:
In this manuscript, the authors investigate different SNPs in vitamin D receptor gene (VDR) in relation to the health of oral cavity of smokers and non-smokers subjects. The results confirm the involvement of genetic and smoke habits in oral health.
This document can be accepted although some points need to be clarified:
1) Clarify the methodology used in the analysis of samples and results, there are some shortcomings in the descriptions.
2) Please describe more in detail the specific components included in the tables 1-4, in order to help understanding the sentence included in the text.
Thank you very much for this suggestion. Beneath the tables are abbreviations for all statistical abbreviations. Additionally, we changed all figures to better resolution, and modified conclusions to fit better our results.
3) Add a reference for sentence:
- line 218: Meta-analysis of 30 studies suggested that this variant was significantly associated with the susceptibility to periodontitis under dominant genetic model in the overall population.
Thank you for this comment – the reference has been added – line 218.
4) Contextualize, more clearly, people origin differences, highlighted in other studies.
Thank you for this comment – clearer description has been added – lines 227-230.
5) check the formatting of the text, and I suggest enter the name of the SNPs in the same way (or rs7975232 or ApaI, rs1544410 or BsmI, and rs2228570 or FokI)
Thank you for this comment – the format has been changed to rs numbers.
6) “Periodontal disease is unconditionally associated with care for oral hygiene and the use of various types of psychoactive substances, especially inhaled - which includes smoking, among others. Such addiction affects not only the health of the oral cavity, but also the transmission of diseases by droplet, important in the light of recent events associated with the COVID-19/ SARS-CoV pandemic.” the sentence seems detached from the rest of the text, could be eliminated or should be inserted in a context and with specific reference.
Thank you for this comment – this sentence was removed.
Round 2
Reviewer 1 Report
The second paragraph of the introduction should start with "Vitamin D Receptor (VDR)..."
The figures and figure legends should be in the same pages.
Author Response
ANSWER
Dear Reviewer,
We would like to thank you for your valuable comments on the article. Below you will find our reply to your review. All changes are with a description or a comment and changes have been made to the manuscript (track changes in the tracking group on the review tab).
Reviewer 1
Comments and Suggestions for Authors
The second paragraph of the introduction should start with “Vitamin D Receptor (VDR)…”
Thank you for this comment – the paragraph has been changed – from line 55.
The figures and figure legends should be in the same pages.
Thank you for this suggestion – figures and its legends are on the same pages.
Reviewer 2 Report
Dear authors, The manuscript improve considerably, thank you for responding the suggestions, now I recommend this manuscript for publication
Author Response
Thank you.
Reviewer 3 Report
Thanks for making the required changes.
Author Response
Thank you.